# FOXM1, MEK, and CDK4/6: New Targets for Malignant Peripheral Nerve Sheath Tumor Therapy

**DOI:** 10.3390/ijms241713596

**Published:** 2023-09-02

**Authors:** Ellen Voigt, Dawn E. Quelle

**Affiliations:** 1Cancer Biology Graduate Program, University of Iowa, Iowa City, IA 52242, USA; ellen-voigt@uiowa.edu; 2Medical Scientist Training Program, Carver College of Medicine, University of Iowa, Iowa City, IA 52242, USA; 3Holden Comprehensive Cancer Center, University of Iowa, Iowa City, IA 52242, USA; 4Department of Neuroscience and Pharmacology, Carver College of Medicine, University of Iowa, Iowa City, IA 52242, USA; 5Department of Pathology, Carver College of Medicine, University of Iowa, Iowa City, IA 52242, USA

**Keywords:** MPNST, FOXM1, MEK, CDK4/6, targeted therapy, sarcoma

## Abstract

Malignant peripheral nerve sheath tumors (MPNSTs) are deadly sarcomas, which desperately need effective therapies. Half of all MPNSTs arise in patients with neurofibromatosis type I (NF1), a common inherited disease. NF1 patients can develop benign lesions called plexiform neurofibromas (PNFs), often in adolescence, and over time, some PNFs, but not all, will transform into MPNSTs. A deeper understanding of the molecular and genetic alterations driving PNF–MPNST transformation will guide development of more targeted and effective treatments for these patients. This review focuses on an oncogenic transcription factor, FOXM1, which is a powerful oncogene in other cancers but little studied in MPNSTs. Elevated expression of FOXM1 was seen in patient MPNSTs and correlated with poor survival, but otherwise, its role in the disease is unknown. We discuss what is known about FOXM1 in MPNSTs relative to other cancers and how FOXM1 may be regulated by and/or regulate the most commonly altered players in MPNSTs, particularly in the MEK and CDK4/6 kinase pathways. We conclude by considering FOXM1, MEK, and CDK4/6 as new, clinically relevant targets for MPNST therapy.

## 1. Introduction

Malignant peripheral nerve sheath tumors (MPNSTs) are a deadly form of sarcoma, and they originate from the Schwann cells in the myelinating nerve sheath [1]. Half of these tumors occur spontaneously in the population, but the other half occur in patients with a hereditary condition called neurofibromatosis type I (NF1). All MPNSTs are initiated by loss of the neurofibromin 1 tumor suppressor gene (*NF1*), resulting in hyperactive RAS signaling [2]. However, patients with NF1 first present with non-cancerous lesions called benign plexiform neurofibromas (PNFs), which have a 30% lifetime risk of transforming into malignant MPNSTs (Figure 1) [2]. More research into the molecular drivers of transformation is desperately needed for MPNST patients, since effective therapies are lacking [3]. Standard treatment is surgical resection, but in many cases, the MPNSTs are inoperable or cannot be fully resected due to their large size and/or location [2,4,5]. With incomplete local resection, radiotherapy and chemotherapy are used for local control of disease but offer no survival benefit [6,7,8]. The overall 5-year survival for MPNST patients is 20–35%, which drops significantly for unresectable or metastatic disease [2,5,9].

To create targeted therapeutics for improved treatment of MPNSTs, we must better understand what triggers the transformation of these benign lesions into malignant tumors. Benign PNFs have homozygous *NF1* loss, typically through loss of heterozygosity via genetic mutation, without other known molecular alterations [10,11]. They usually slow or stop growing and do not display malignant behaviors [2,12]. Recently, an intermediate stage of benign lesion with more potential to transform into MPNST has been identified and termed atypical neurofibromatous neoplasms of uncertain biological potential (ANNUBPs) [13,14]. After *NF1* loss, nearly all ANNUBPs are characterized by the deletion or silencing of the *INK4a/ARF* (originally called *CDKN2A*) tumor suppressor locus [15,16,17,18], which is inactivated in the majority of human cancers [15,19,20].

An increasing number of additional molecular alterations currently discriminate fully transformed human MPNSTs from ANNUBPs, although that understanding is incomplete and represents a major gap in the field [21,22,23]. While many changes have been observed in MPNSTs by our lab and others, this review is centered on the interplay between FOXM1, a little studied player in this disease, and the more commonly altered genes and pathways, which have documented contributions to MPNST pathogenesis (Figure 1). FOXM1, an important oncoprotein and promising target in other cancers [24,25], is highly overexpressed and a marker of poor survival in patient MPNSTs [26], but its role in driving the disease is otherwise not known. In addition to loss of *NF1* and *INK4a/ARF*, other key players in MPNST include the globally important tumor suppressor, p53, and two epigenetic regulators in the polycomb repressive complex 2 (PRC2), EED and SUZ12, all of which are lost in MPNSTs [16,27,28]. Prominent oncoproteins in addition to FOXM1, which are often overexpressed in MPNSTs, include cyclin dependent kinase 4 (CDK4), the Yes-associated protein 1 (YAP1, hereafter called YAP), and a Rab-like GTPase called RABL6A [3,16,26]. How these factors are altered in MPNSTs and cross-talk with each other, as well as with FOXM1, is discussed below.

## 2. FOXM1 in Cancer and MPNST

Forkhead box protein M1, a member of the forkhead box (Fox) transcription factor family, controls gene expression during embryonic development and maintains cell homeostasis throughout life by regulating essential biological processes, such as cell cycle progression, differentiation, and apoptosis, among others [24]. FOXM1 is a ubiquitously expressed protein in tissues, whose levels and activity fluctuate during the cell cycle. Its activation during G1/S and G2/M phases enable it to transcribe genes needed for cell cycle progression, including proteins required for DNA replication [29]. FOXM1 is a winged helix transcription factor, defined by its DNA binding domain of three α-helices, three β-sheets, and two wings/loops bookending the last β-sheet [24]. It has ten exons, which can be spliced into four different isoforms: FOXM1a, b, c, d [24]. FOXM1b and FOXM1c are the transcriptionally active forms of the protein and the most often overexpressed in cancers [30,31].

FOXM1 has been recognized as a powerful oncogene and was named the molecule of the year in 2010 for its promise as a cancer therapeutic target [30]. It is essential for growth and survival in many cancers, including melanoma, lung, ovarian, breast, prostate cancers, and numerous sarcomas [29,32,33,34,35,36]. Relevant to MPNST, a tumor initiated by Ras hyperactivation, FOXM1 is required for growth in other RAS-driven tumors, such as hepatocellular carcinoma, colorectal cancer, and pancreatic ductal adenocarcinoma [37,38,39,40,41]. Mechanistically, FOXM1 has been implicated in many important oncogenic processes, such as angiogenesis [42,43], metastasis [44,45,46], stem cell maintenance and de-differentiation [47,48], DNA damage response/repair [49], and drug resistance [50,51].

Only one published study has measured the FOXM1 gene and protein levels in MPNSTs. Yu et al. performed array-based comparative genomic hybridization on a large number of patient MPNSTs to uncover survival-associated biomarkers [26]. They showed that increased copy number of the *FOXM1* gene (chromosome 12p13.33) occurred in 29% of MPNSTs and correlated with worse patient survival. Concordantly, elevated protein expression of FOXM1 through immunohistochemical (IHC) staining was a significant independent predictor for poor survival. In an RNA sequencing (RNA-Seq) dataset from our lab used to compare the transcriptomes of patient-matched PNF/ANNUBPs and MPNSTs, *FOXM1* mRNA and transcript levels of key transcriptional targets (e.g., *AURKB*, *BIRC5*, *CENPA*, *CCNB1*, *CDK1*) [52,53] were significantly increased in MPNSTs [54]. The IHC of the same tumor samples likewise revealed robust upregulation of the FOXM1 protein in MPNSTs relative to the benign precursors [55].

Beyond these correlative observations, however, the effect of FOXM1 on PNF transformation or MPNST tumor progression has not been evaluated.

## 3. Upstream Regulators of FOXM1

### 3.1. FOXM1 Upregulation through INK4a/ARF Loss

During the transition into the intermediate stage of premalignancy called ANNUBP, most PNFs lose the *INK4a/ARF* locus, which researchers have shown is required for this transformation [14,17,18]. *INK4a/ARF* encodes two powerful tumor suppressors: p16^INK4a^ and the alternative reading frame product called ARF (p14 in human, p19 in mouse) [20,56]. *INK4a* and *ARF* have unique first exons spliced to shared exons 2 and 3; consequently, their open reading frames and encoded proteins are entirely distinct [20]. p16^INK4a^ is so named because it is a specific inhibitor of CDK4 and CDK6 [57], two nearly identical cyclin dependent kinases, which inactivate the retinoblastoma (RB1) tumor suppressor [20]. As such, loss of *INK4a* heightens CDK4/6 activity and increases G1/S cell cycle progression [20]. The second protein product from this locus, ARF, is a small, highly charged protein, which prevents cancer through activation of p53, as well as numerous p53-independent mechanisms due to its interactions with over forty proteins, one of which is FOXM1 [58].

ARF binds directly to FOXM1b at the C-terminus within its transcriptional activation domain, which inhibits FOXM1 transcriptional activity by mobilizing the protein into the nucleolus (Figure 2) [37,59]. In 2011, Park et al. examined the effects of liver-specific Foxm1b expression in mice lacking *Arf* and found that dysregulated Foxm1b expression in *Arf+/−* and *Arf−/−* settings, but not in *Arf+/+* wild-type mice, promoted the metastasis of hepatocellular carcinoma (HCC) [44]. Excitingly, a synthetic p19Arf peptide (amino acids 26 to 44) was sufficient to sequester Foxm1b in nucleoli [37], effectively inhibiting the primary and metastatic growth of *Foxm1b* transgenic; *Arf−/−* liver cancer cells without affecting normal hepatocytes [44,60]. ARF may also indirectly impair FOXM1 expression by upregulating miR-34a [61], which is one of many miRNAs able to downregulate FOXM1 [24]. Conversely, FOXM1c can downregulate ARF expression and impair p53 activation by promoting expression of the polycomb group protein Bmi-1, thereby blocking cell senescence and promoting proliferation [62].

While the above studies report data for either FOXM1b or FOXM1c, since either isoform can be the predominant protein within a particular cancer, it is expected that both proteins will have similar biochemical activities. Thus, when ARF is upregulated in response to oncogenic stress, such as Ras activation in benign PNFs, its negative regulation of FOXM1 would block its function and enforce senescence. Conversely, loss of the ARF in ANNUBPs and MPNSTs would be predicted to heighten FOXM1 activity and promote tumorigenesis.

The functional relationship between p16^INK4a^ and FOXM1 is less direct but nonetheless impactful. FOXM1 represses transcription of the FOXO1 transcription factor, which normally transactivates the genes encoding several CDK inhibitory proteins p27^KIP1^, p15^INK4b^, and p16^INK4a^ [63,64]. As such, knockdown of FOXM1 increases the expression of p16^INK4a^ and other CDK inhibitors, whereas FOXM1 overexpression suppresses their transcription and causes elevated activity of tumor-promoting CDK2 and CDK4/6 [24,63]. As *INK4a/ARF* loss is central to the oncogenesis of MPNST, these functional interactions with FOXM1 suggest that FOXM1 may also be an important driver in this cancer (Figure 2).

### 3.2. FOXM1 Control by MEK and CDK4/6 

In PNFs, the inciting mutation is the loss of heterozygosity of *NF1*, which produces the protein neurofibromin. Neurofibromin is a Ras-GTPase, whose loss results in inefficient catalysis of the active GTP-bound Ras to inactive GDP-bound Ras, thereby hyperactivating Ras signaling [12]. A major downstream mediator of Ras is the MEK-ERK1/2 kinase cascade, which can act both directly and indirectly on FOXM1 to promote its activation (Figure 2). ERK1/2 phosphorylates FOXM1c at two different serine residues—S331 in the DNA binding domain and S704 in the transcriptional activation domain—which promotes the translocation of FOXM1 into the nucleus and its transcriptional activation (Figure 3) [65,66].

Activated MEK-ERK1/2 signaling can also increase FOXM1 protein levels indirectly by promoting the expression and activation of cyclin D-CDK4/6 kinases, whose phosphorylation of FOXM1 effectively stabilizes and activates the protein [67,68]. Anders et al., showed that cyclin D-CDK4/6 kinases can phosphorylate many S and T residues in FOXM1, including those depicted in Figure 3 [67]. Single phosphorylation events were not sufficient to promote FOXM1 transcriptional activity, but combined phosphorylation of five to seven sites in the FOXM1 C-terminus (T600, T611, T620, T627, S638, S672, and S704) effectively increased FOXM1 activity. Those findings were consistent with prior work showing that T600, T611, and T638 are critical for FOXM1 function [69]. While CDK4/6-mediated phosphorylation of residues in the FOXM1 N-terminus and DNA binding domain (S4, S35, S451, S489, S508, T510, and S522) do not lead to transactivation, they provide critical contributions to full FOXM1 activation [67]. Biologically, the activation of FOXM1 by CDK4/6 is critical for CDK4/6-mediated cell cycle entry, suppressing the levels of reactive oxygen species (ROS), and protecting cancer cells (breast, melanoma, sarcoma) from senescence [67].

As indicated above, FOXM1 is a highly phosphorylated protein [24,67]. Some phosphorylation events facilitate FOXM1 association with CBP/p300, a transcriptional coactivator, which supports FOXM1-mediated gene transcription [70]. Phosphorylation of FOXM1 at other residues regulates its degradation; some stabilize it, while others mediate its ubiquitination and degradation, as reviewed in detail elsewhere [24]. In addition to being phosphorylated by cyclin D-CDK4/6 in late G1/early S phase (see Figure 3), this includes phosphorylation at the same or additional serine/threonine residues by cyclin E-CDK2 during the G2/M phase, cyclin A-CDK2 during the G2/M phase, and cyclin B-CDK1 during the G2 phase [30,69,71,72,73]. In non-transformed cells, FOXM1 is typically hypo-phosphorylated during the early G1 phase and minimally active, increasingly phosphorylated during the late G1 and S phase to become more active, followed by the highest levels of phosphorylation and transcriptional activation through G2/M before becoming dephosphorylated at the end of the M phase [30]. In MPNST, where *INK4a* is lost and CDK4 protein is overexpressed, CDK4/6 complexes are hyperactive and would be expected to aberrantly phosphorylate FOXM1, thereby increasing its stability and inappropriately increasing its transcriptional activity during the G1/S phase [67]. Under such circumstances, FOXM1 would be expected to drive MPNST cell proliferation and prevent cells from undergoing senescence.

In sum, there is abundant evidence for the upregulation of MEK and CDK4/6 in MPNST [3]. Both kinases promote increased FOXM1 protein levels, FOXM1 stability, and FOXM1 transcriptional activity. Together, those observations heighten the likelihood that FOXM1 is a major contributor to MPNST pathogenesis.

### 3.3. Opposing Transcriptional Regulation of FOXM1 by p53 and YAP

The p53 transcription factor is the most frequently inactivated tumor suppressor in human cancers [74]. Genetically, *TP53* is mutated or deleted in about 55% of human cancers, while p53 signaling is likely impaired in the remaining tumors [61]. Mutations of *TP53* are either loss-of-function (missense, deletion) or gain-of-function in nature, with the latter often associated with heightened stability and an altered transcriptional program, which is pro-oncogenic [74]. In human MPNSTs, the percentage of tumors bearing inactivated *TP53* ranges from approximately 14% to 50%, depending on the study [26,75,76,77]. In a comprehensive analysis by Verdijk et al., 24% of MPNSTs had a loss of the *TP53* gene in their study of 145 cases as compared to a 14% loss of *TP53* in 411 cases in the literature at that time [76]. Nonetheless, loss of p53 tumor suppressive activity can clearly drive the disease, as shown by the number of p53-deficient mouse tumor models, which develop MPNSTs [16,61,78].

Whether or not p53 may normally suppress MPNSTs, at least in part, by dampening FOXM1 expression and activity has not been tested. However, several lines of evidence in other models suggest that may be the case. In several cell types, it has been shown that *FOXM1* transcription can be repressed by p53 (Figure 2) [79,80]. In breast cancer, p53 represses *FOXM1* transcription through association with E2F1 at the *FOXM1* promoter [79]. However, this effect on FOXM1 appears to be context-dependent, as p53 is reported to activate transcription of the *FoxM1* gene during liver regeneration in mice [81]. In addition to transcriptional control, wild-type p53 is reported to negatively regulate *FOXM1* mRNA [82]. This repression of FOXM1 by p53 is partially dependent on p21 and RB, and in some cells, it has been shown to be independently repressed by p21 [82]. In the development of a gastroesophageal cancer organoid model through the dual CRISPR/Cas9 knockout of *Ink4a/Arf* and *Tp53*, Zhao et al. showed that FOXM1 was upregulated and mediated pro-tumorigenic epigenetic changes [83].

YAP is an oncogenic transcriptional coactivator, which is turned off in non-transformed cells by tumor suppressive Hippo/LATS1/2 signaling and conversely upregulated in tumor cells [84]. Multiple points of cross-talk exist between YAP and p53, which are important to both normal cell control and tumorigenesis, the nuances and complexities of which are comprehensively reviewed elsewhere [84,85]. Of interest here are the connections to FOXM1. Nuclear YAP, in association with the TEAD transcription factor, can bind directly to the *FOXM1* promoter and stimulate its expression (Figure 2) [86]. Wild-type p53, which can repress *FOXM1* transcription through p21/RB/E2F mechanisms noted above, can also downregulate *FOXM1* mRNA levels by inhibiting YAP. It does so by increasing the expression of 14-3-3σ or PTPN14, both of which retain YAP in the cytoplasm, where it is unable to activate tumor-promoting transcriptional programs [87,88,89].

Although *FOXM1* is just one of many YAP gene targets, it appears to be a critical mediator of YAP-dependent tumorigenesis. In a transgenic mouse model of hepatocellular carcinoma driven by constitutively active YAP (YAP^S127A^), inhibition of FOXM1 with thiostrepton blocked the YAP-induced chromosomal instability phenotype [90]. In soft tissue sarcomas, which include MPNSTs, the Hippo pathway is frequently dysregulated, leading to increased expression of YAP and its transcriptional coactivator TAZ [91,92]. Not only do elevated YAP and TAZ independently predict worse overall and progression-free survival, especially in the absence of p53 [92]; YAP’s upregulation of FOXM1 was found to be necessary for cell proliferation and tumorigenesis in a subset of sarcomas [91].

### 3.4. RABL6A and FOXM1

Our lab discovered an unusually large Rab-like GTPase called RABL6A, which promotes cancer [3,54,92,93,94,95,96]. RABL6A overexpression is a marker of poor survival in several cancers and is required for growth in pancreatic, breast, and esophageal cancers, among others [97,98,99,100,101,102]. Using a tissue microarray of patient-matched PNFs, ANNUBPs, and MPNSTs, RABL6A protein expression was found to be greatly increased in MPNSTs compared to low/undetectable levels in PNFs and moderately higher but still low levels in ANNUBPs [93]. Those findings prompted analyses of RABL6A silencing in MPNST cell lines, which showed that RABL6A loss causes significant cell death in vitro [93]. In agreement, a genetic knockout of *Rabl6* in mice delayed tumor growth in a de novo model of MPNST induced by *Nf1-Ink4a/Arf* deletion [54]. Together, these data revealed that MPNST cell survival and tumor growth are driven by RABL6A, but how it cooperates with *Ink4a* and *Arf loss* to promote MPNST progression is not known.

A connection between RABL6A and FOXM1 has yet to be explored, but a functional link is likely, since they engage many of the same factors and pathways shown in Figure 2. Most notably, RABL6A was originally discovered based on its ability to bind to ARF, and its expression in a mouse model of pancreatic neuroendocrine tumors (pNETs) was associated with p19Arf downregulation [95,98]. In both MPNST and pNET human cell lines, the silencing of *RABL6A* caused robust upregulation of the CDK inhibitors p27^KIP1^ and p21^CIP1^, leading to inhibition of CDK4/6-mediated RB1 phosphorylation and consequent cell cycle arrest and death [93,98]. RABL6A was also found to promote Schwann cell proliferation and abrogate their senescence by inhibiting RB1 [103]. A role for RABL6A in Schwann cell biology is relevant to PNF and MPNST development, since Schwann cells are the non-transformed precursors of those tumors. In addition, RABL6A is a critical activator of MEK-ERK signaling [99,104], and its overexpression in patient MPNSTs is significantly associated with an activated Ras-MEK pathway [105]. Finally, in biomarker analyses of 163 sarcomas representing many different histological types, RABL6A expression was positively correlated with high levels of p53 (likely mutated) and YAP [92].

The above commonalities in RABL6A and FOXM1 pathways suggest that FOXM1 expression and activity may be upregulated by RABL6A in cells. It will be interesting to examine that possibility and to determine whether FOXM1 and RABL6A act cooperatively to promote tumorigenesis, not only in driving PNF to MPNST transformation but in the malignant progression of other sarcoma types and solid tumors, where both proteins are highly expressed.

## 4. Downstream Targets of FOXM1

### 4.1. FOXO1

FOXO1 belongs to the large forkhead box transcription factor superfamily, just like FOXM1, but it forms part of a subclass along with FOXO3, FOXO4, and FOXO6 [106]. Like the other Fox family members, these transcription factors are important in the development of normal tissues and in stem cell maintenance/differentiation [107]. Unlike FOXM1, the FOXO transcription factors are tumor suppressors and are often inactivated in cancers [107,108]. FOXO1 is of particular interest in MPNST because it activates the transcription of multiple cell cycle inhibitory proteins, including p27^KIP1^, p15^INK4b^, and p16^INK4a^ [63,64]. FOXO1 has also been shown to inactivate metastasis programs. FOXO1 overexpression inversely correlates with genes, which promote the epithelial–mesenchymal transition (EMT) [109]. Indeed, FOXO1 directly represses the transcription of *ZEB2*, itself a transcription factor, which induces EMT and thereby increases tumor metastasis [109]. Importantly, FOXM1 and FOXO expressions are also inversely correlated in cancers [110]. Chand et al. showed that FOXM1 represses *FOXO1* transcriptionally through recruitment of RB1 and DNMT3B and increased methylation at the FOXO1 promoter in hepatocellular carcinoma cells [63]. If the same is true in MPNST, the elevated expression of FOXM1 would be expected to suppress *FOXO1* transcription, thus downregulating expression of the FOXO1 protein and impairing its induction of CDK inhibitors (Figure 4).

### 4.2. Multiple Modes of RB1 Regulation by FOXM1 

FOXM1 controls RB1 in two opposing ways, either to turn off the canonical cell cycle inhibitory function of RB1 to enhance cell proliferation or to engage RB1 within repressive transcriptional complexes, which ultimately promote tumor metastasis.

In the first situation, recall that cyclin D-CDK4/6 and cyclin E-CDK2 kinase complexes phosphorylate and inactivate the RB1 tumor suppressor during the G1-to-S transition, thereby preventing its association with E2F transcription factors [111]. E2F is then free to transcribe the genes needed for S phase entry and progression. In many cancers including MPNST, RB1 is kept in a hyperphosphorylated state, such that the cell cycle progresses unchecked, and cells proliferate uncontrollably [93]. Since E2F activates the transcription of *FOXM1* [79], aberrant RB1 phosphorylation also results in abnormally high expression of the FOXM1 protein. Upregulated FOXM1 can then amplify RB1 hyperphosphorylation through its inhibition of FOXO1, as described above, as this causes downregulation of CDK inhibitors and consequently elevated activity of CDK4/6 and CDK2 holoenzymes [63].

With regard to the second scenario, FOXM1 is generally regarded as a master activator of genes promoting cell proliferation, survival, metastasis, and drug resistance [45]. However, FOXM1 can also repress gene expression [112]. Recent studies in mouse breast cancer models showed that FoxM1 binds directly to the RB1 protein and forms repressive transcriptional complexes, which suppress the *Pten* tumor suppressor gene and are essential for tumor cell plasticity and metastasis (Figure 4) [35]. This ‘pro-metastatic’ role of RB1 when bound to FOXM1 flies against the conventional view of RB1 as a tumor suppressor. Nonetheless, the study by Kopanja et al., effectively demonstrated that a FOXM1 mutant unable to bind to RB1 was deficient in supporting breast cancer de-differentiation and metastasis [35]. Human tumors, which already display PTEN downregulation, such as the majority of MPNSTs [113,114], may contain a higher fraction of FOXM1–RB1 complexes, which enhance their metastatic potential.

### 4.3. EZH2 Cooperation with FOXM1

The polycomb repressive complex 2 (PRC2) is a large multi-protein complex composed of three main factors. The enhancer of zeste homolog 2 (EZH2) is a histone methyltransferase and the catalytic subunit of PRC2, while the suppressor of zeste 12 (SUZ12) and embryonic ectoderm development (EED) proteins are core regulatory and scaffolding subunits of the holoenzyme [115]. PRC2 is a global regulator of transcription, whose dysregulation in cancer is complex and associated with metastasis, chemotherapy resistance, and poor prognosis [116]. While PRC2 is oncogenic in many cancers, the majority of MPNSTs (and some other tumor types) display loss-of-function mutations in *SUZ12* and/or *EED*, which inactivate PRC2 and cause broad epigenetic dysregulation [16,27,116,117]. Analyses of patient MPNSTs with or without PRC2 loss showed that its inactivation caused increased histone post-translational modifications associated with active transcription and loss of certain repressive histone modifications [118]. The resulting epigenome correlated with proteomic changes reflecting tumor progression and immune evasion. Most recently, Brockman et al. found that PRC2 inactivation (due to *Suz12* or *Eed* loss in mouse MPNST models) induced the expression of matrix-remodeling enzymes (matrix metalloproteinases, or MMPs) and increased lung metastasis [117]. Analysis of clinical samples similarly revealed increased metastatic disease and decreased survival in patients, whose MPNSTs displayed PRC2 loss.

Several studies have demonstrated that FOXM1 cooperates with EZH2 to promote tumor growth, metastasis, and radioresistance (Figure 4) [119,120,121]. In glioblastoma stem cells, FOXM1 is phosphorylated by and associates with maternal embryonic leucine-zipper kinase (MELK), and both events are necessary for a MELK–FOXM1 protein complex to bind the *EZH2* promoter and stimulate its transcription [120]. MELK and FOXM1 were found to be the predominant activators of *EZH2* transcription in these tumor stem cells, and MELK–FOXM1–EZH2 signaling mediated cellular resistance to irradiation. In prostate cancer, FOXM1 is highly expressed, and its regulation of EZH2 is essential for tumor cell proliferation and progression [119]. While those investigations focused only on EZH2 rather than PRC2, Mahara et al. examined EZH2 regulation in triple-negative breast cancer relative to PRC2 functionality [121]. They found that hypoxia causes PRC2 inactivation due to HIF-1α suppression of *SUZ12* and *EED*, thus liberating EZH2, which then forms a complex with FOXM1 and transcriptionally induces the expression of *MMP* genes (Figure 4) [121]. The hypoxia-induced shift of EZH2 from PRC2 into FOXM1 complexes promoted tumor cell invasion. Based on those findings, we speculate that the genetic loss of *SUZ12* and *EED* in the majority of MPNSTs may similarly enhance the formation of EZH2–FOXM1 complexes, which would then drive MMP expression and the metastatic phenotype.

### 4.4. YAP/TEAD Cooperation with FOXM1

In multiple sarcoma types, YAP expression is elevated through disruptions in the Hippo signaling pathway, and it is known to promote *FOXM1* gene expression [29,86,91,92]. Interestingly, FOXM1 physically interacts with YAP/TEAD complexes in sarcoma and hepatocellular carcinoma to alter the transcription of genes necessary for promoting proliferation and blocking apoptosis [90,91]. In hepatocellular carcinoma, this trimeric complex containing FOXM1 was also associated with increased expression of genes responsible for aneuploidy and chromatin instability—features known to enhance tumorigenic potential [90]. In breast cancer models, FOXM1 blocked the phosphorylation of YAP at S127, thereby enhancing YAP’s nuclear localization and ability to promote the transcription of proliferation, migration, and cell stemness genes [122]. Using integrated omics and drug screening approaches, Nilsson et al. found that FOXM1–YAP signaling drove resistance to tyrosine kinase inhibitor therapy targeting the epidermal growth factor receptor (EGFR) by upregulating genes encoding spindle assembly checkpoint proteins, such as polo-like kinase 1 (PLK1), aurora kinases, and survivin [123].

The above data reveal that FOXM1 is a biologically relevant activator of YAP. Since YAP also transactivates *FOXM1*, the cumulative findings establish a FOXM1–YAP positive feedback loop, where they can activate each other and drive oncogenesis. A recent transposon mutagenesis guided CRISPR screen in immortalized Schwann cells strongly implicates YAP hyperactivation in the transformation of PNFs, providing further evidence for a key role of the FOXM1–YAP axis in MPNST development [124].

## 5. Targeting the MEK–CDK4/6–FOXM1 Axis to Treat MPNST

### 5.1. Relevance of MEK and CDK4/6 Inhibitor Therapies

Effective therapeutic options for unresectable MPNSTs are woefully lacking, correlating with a dismal 5-year survival rate of 20–35% for these patients [92]. The challenges of performing clinical trials in a rare cancer have certainly delayed progress, but so has the aggressive nature and limited responsiveness of MPNSTs to standard chemotherapeutics. We recently sought to identify new drug combinations, which would have sustained activity against MPNSTs. We began by querying the Connectivity Map (C-Map) database [125] using transcriptomes gathered from patient MPNSTs [105]. Consistent with molecular data from patient tumors, which defined hyperactivated MEK and CDK4/6 as hallmark drivers of MPNSTs, small molecule drugs targeting MEK and CDK4/6 were among the top drug candidates identified. We previously found that CDK4/6 inhibitor monotherapy had excellent anti-tumor effects against de novo MPNSTs in mice, but drug resistance occurred rapidly [93]. In the most recent study, MEK inhibitors alone were ineffective, but low-dose combinations of a MEK inhibitor (mirdametinib) and CDK4/6 inhibitor (palbociclib) acted synergistically in causing remarkable tumor regression and improved survival in immune competent mice bearing MPNSTs [105]. Excitingly, dual MEK–CDK4/6 inhibition induced an anti-tumor immune response, which sensitized MPNSTs to immune checkpoint inhibitor therapy using anti-PD-L1 (programmed death-ligand 1) therapy, with about 10% of mice showing cure with long-term treatment.

The study by Kohlmeyer and Lingo et al. [105] revealed a high potential for MEK–CDK4/6 inhibitor therapy, especially when combined with immunotherapy, to induce sustained tumor regression and better survival in MPNST patients. On the other hand, caution is warranted, since all MEK–CDK4/6 inhibitor treated tumors eventually became resistant during continued therapy, as did most of the tumors given the MEK–CDK4/6–PD-L1 inhibitor triple therapy [105]. Those results firmly established that one or more mechanisms in the tumors are mediating treatment resistance. Many possibilities exist, but as discussed throughout this review and below, upregulation of FOXM1 is a logical potential culprit. If so, it would make sense to include FOXM1 inhibitors in MPNST targeted therapies employing MEK and CDK4/6 inhibitors.

### 5.2. Targeting FOXM1 in MPNST

There is broad interest in targeting FOXM1 in cancer therapy, and we prioritized three reasons for doing so in MPNST. First, FOXM1 is a highly oncogenic protein, whose expression in MPNSTs is associated with poor patient survival [26]. While its role in this disease has not been sufficiently investigated, it is well known that FOXM1 is required for the proliferation, survival, and metastasis of many other cancer types [24,25]. Second, dysregulated transcription factors in cancer, such as FOXM1, orchestrate impactful alterations in gene expression programs and biological processes, which drive tumor pathogenesis. Inhibition of such ‘master regulators’ would therefore be expected to have wide and potentially sustained tumor suppressive effects. Third, FOXM1 mediates tumor cell resistance to irradiation [120], chemotherapies [126,127], and targeted therapeutics, including PI3K inhibitors [128,129], EGFR inhibitors [123], and CDK4/6 inhibitors [67,129], among others. Kopanja et al. noted that CDK4/6 inhibitors, such as palbociclib, not only activate RB1 but also decrease the levels of FOXM1 [35,67]. They speculated that the mechanisms leading to FOXM1 accumulation may contribute to palbociclib resistance in RB1-positive breast tumors. Interestingly, in bladder cancer, the opposite trend was observed. Specifically, high levels of FOXM1 conferred increased sensitivity to CDK4/6 inhibitors, which was independent of RB1 status, and the treatment reduced phosphorylated FOXM1 [130]. Tumor type and RB1 context may affect exactly how FOXM1 expression influences CDK4/6 inhibitor efficacy, meriting further investigation, but there is growing evidence that FOXM1 plays a key role in determining tumor cell responsiveness and resistance to CDK4/6 targeting.

While neither CDK4/6 nor MEK inhibitors are currently approved for treating MPNSTs, patients with inoperable PNFs are given MEK inhibitors to slow growth and even shrink tumors [131,132]. In breast cancer, there is evidence that MEK-activated FOXM1 mediates resistance to lapatinib, a dual EGFR/HER2 tyrosine kinase inhibitor [133]. This suggests that FOXM1 upregulation could mediate resistance to MEK inhibitors, which we speculate might have already occurred in MPNSTs arising from PNFs that were treated with MEK inhibitors. If FOXM1 upregulation in MPNSTs does mediate acquired resistance to CDK4/6 and/or MEK inhibition, pharmacologically blocking FOXM1 activity in combination with drugs targeting MEK and CDK4/6 could be highly effective in achieving sustained MPNST regression. Indeed, recent studies in ER-positive breast cancer models demonstrated that low doses of novel FOXM1 inhibitors acted synergistically with low doses of CDK4/6 inhibitors (abemaciclib, palbociclib, or ribociclib) to efficiently suppress tumor cell growth [134].

So, what agents are available to inhibit FOXM1 in cancer? Early studies revealed that a cell-penetrating ARF inhibitory peptide effectively blocked FOXM1 activity in cultured cancer cells and mice [60,135], but the pharmacokinetics of such peptides are not suitable for clinical use. FOXM1 is a transcription factor, and for many years, efforts to develop drugs, which effectively target such proteins, were largely unsuccessful. However, recently, there have been technological advances in drug development along with novel approaches to abrogating transcription factor activity, resulting in a number of promising pharmaceuticals for transcription factor inhibition [136]. For FOXM1, several direct inhibitors have been reported. Two structurally similar thiazole antibiotics produced by *Streptomyces* species, siomycin A and thiostrepton, were identified through cell-based screens as effective FOXM1 inhibitors, which had significant anti-cancer activity [137,138,139]. Using an affinity-tagged thiostrepton analog and various biophysical analyses, Hegde et al. determined that thiostrepton binds both FOXM1b and FOXM1c directly and prevents them from associating with DNA—not only radiolabeled DNA in cell-free assays but also specific genomic promoters within cells [140]. In breast cancer studies, thiostrepton was found to be effective at inhibiting cell migration in vitro and metastasis in vivo; unfortunately, the poor stability and solubility of this natural product make future application in the clinic challenging [141]. Clinical use of thiostrepton is also limited by its lack of specificity, as it has other important molecular targets, namely the proteasome and mitochondrial translation machinery.

Another group of small molecules named FDI-(1-16) was identified by Gormally et al. in a drug screen with 54,211 compounds [142]. One of the compounds, FDI-6, was shown to bind FOXM1 and displace the protein from DNA. The bulk RNA-Seq signature from treated cells was consistent with the inhibition of FOXM1. Unfortunately, the in vivo bioavailability of FDI-6 was very poor in mice [143]. Another recently developed inhibitor, STL427944, binds FOXM1 and re-localizes it from the nucleus to the cytoplasm [144]. A third small molecule discovered in a large in silico screen in 2022, called DZY-4, associates with the DNA binding domain of FOXM1 [145]. DZY-4 is predicted in silico to be specific for FOXM1 over five other FOX family proteins, which share similar morphology with FOXM1. In addition, there have been advancements in the development of proteolysis-targeting chimeras (PROTACs), which bring E3 ligase into close proximity to molecular targets, such as FOXM1 [146,147].

More recently developed compounds, which are also able to bind FOXM1, promote its degradation, and inhibit breast cancer proliferation, show greater clinical promise [143], as comprehensively reviewed by Katzenellenbogen et al. [148]. These compounds, representing a new class of synthetic 1,1-diarylethylene mono- and di-amine molecules, bound to FOXM1 with excellent affinity, which correlated with their cellular potencies. Cell-based assays suggested that FOXM1 association with the compounds perturbed FOXM1 conformation, making the protein more susceptible to proteolysis. Importantly, several compounds displayed favorable pharmacokinetic properties at low micromolar doses in vivo and effectively suppressed breast tumor xenograft growth, as well as FOXM1-regulated genes. A few of the inhibitors displayed excellent half-lives and blood levels after subcutaneous administration in mice, while one compound (NB-55) also had good (albeit not outstanding) activity when given orally. Most excitingly, several of these new FOXM1 inhibitors were tested in combination therapies and found to act synergistically with CDK4/6 inhibitors, as well as proteasome inhibitors against ER-positive breast cancer [134]. While the selectivity of these drugs to specifically inhibit FOXM1 has yet to be determined, this set of FOXM1 inhibitors have compelling translational potential to one day be clinically tested in logical combination therapies against cancers driven by FOXM1. This would likely include breast cancer, hepatocellular carcinoma, and prostate cancer, to name a few, and possibly MPNST, if future investigations verify our prediction based on mounting evidence that FOXM1 is a critical mediator of its pathogenesis.

## 6. Conclusions

MPNSTs are highly aggressive and deadly sarcomas, with resection as the only curative option for patients. As half of these tumors originate in patients with neurofibromatosis type 1 and begin as benign PNFs, we must better understand the early genetic and molecular changes in these lesions, which drive transformation. Many of the alterations observed in human tumors have not been fully characterized or evaluated experimentally. This includes the observed overexpression of FOXM1 in MPNSTs, which was found to be predictive of poor patient survival. As FOXM1 is known to regulate or be regulated by many of the frequently altered tumor suppressors and oncogenes in ANNUBPs and MPNSTs, it represents a novel and relevant factor warranting further study in this cancer. In addition, promising new drugs have been developed to target FOXM1 and represent rational candidates for new combination therapies to treat MPNSTs and potentially prevent resistance to other treatments.

## Figures and Tables

**Figure 1 ijms-24-13596-f001:**
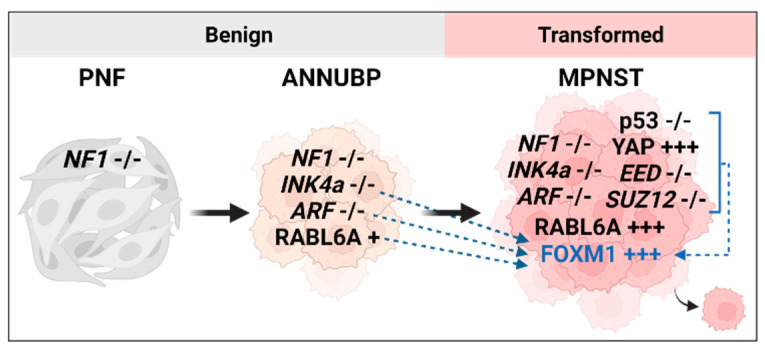
Diagram of known alterations (black, incomplete list) and their proposed interactions with FOXM1 (blue, dashed arrows) to drive MPNST transformation. This figure was created with BioRender.com.

**Figure 2 ijms-24-13596-f002:**
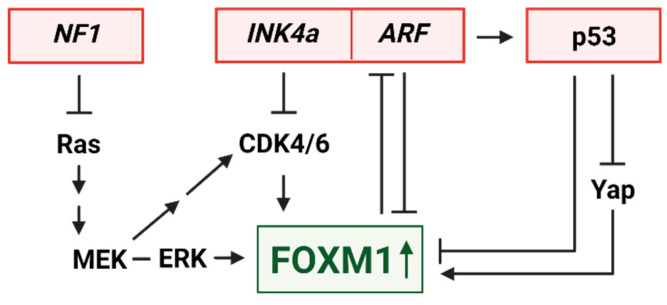
Select upstream regulators of FOXM1, which are key players in MPNSTs. Tumor suppressors, whose inactivation is documented to drive MPNSTs in both NF1-associated and sporadic tumors, are highlighted in red. Oncogenic factors, whose activation promotes MPNST development, are indicated in black text. All displayed factor alterations are known to increase FOXM1 expression in other cancers, as shown by green arrow. Arrows convey activating events, perpendicular bars convey inhibition. This figure was created with BioRender.com.

**Figure 3 ijms-24-13596-f003:**
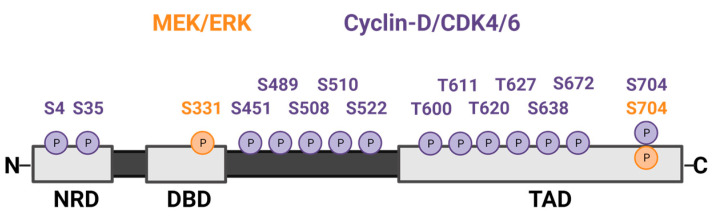
Selected serine (S) and threonine (T) phosphorylation sites on the FOXM1b/c protein mediated by MEK-activated ERK (orange) and CDK4/6 kinases (purple). The protein regions are NRD, negative regulatory domain; DBD, DNA binding domain; TAD, transcriptional activation domain. This figure was created with BioRender.com.

**Figure 4 ijms-24-13596-f004:**
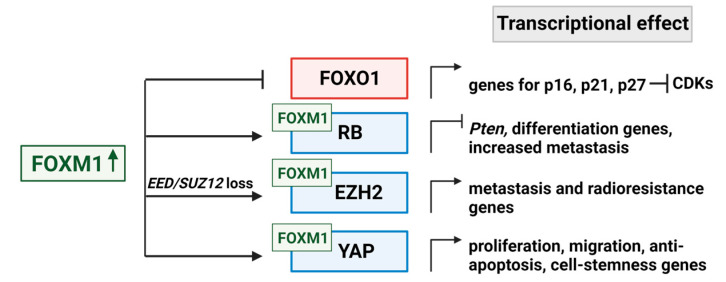
Select transcriptional events downstream of FOXM1 with predicted relevance in MPNST. Increased FOXM1 expression (green arrow) promotes cancer formation primarily through alteration of transcription. FOXM1 can inhibit other transcription factors (FOXO1, shown in red). Alternatively, FOXM1 can bind other proteins to activate or repress transcription (proteins in blue). Arrows convey activating events, perpendicular bars convey inhibition. This figure was created with BioRender.com.

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
