# Peer review of "FOXM1, MEK, and CDK4/6: New Targets for Malignant Peripheral Nerve Sheath Tumor Therapy"

_ijms, 2023, doi:10.3390/ijms241713596_

Round 1

Reviewer 1 Report

It is a comprehensive review about the role of FOXM1 in MPNST. However, there are several questions exist.

1.What is the evidence that FOXM1 is a driver of MPNST rather than biomarker or modulator of chemoresistance?

2. There more than dozen of known FOXM1 inhibitors. What is the logic to describe ARF peptide that fails trials or proteasome inhibitors that non-specifically inhibit FOXM1.

3. The notion that Thiostrepton directly binds to FOXM1 and inhibits its activity is questionable. The problem is that all proteasome inhbitors inhibit FOXM1 via activation of HSP70. But is was never shown that different proteasome inhibitors with different structures bind directly to FOXM1. Therefore, it is more probable then not that general reason for proteasome inhibitors to inhibit FOXM1 is a common up-regulation of HSP70.

Author Response

We thank the reviewers for their time and effort reviewing our manuscript, as well as their constructive comments and questions. Below is a point-by-point response to each comment. We made important revisions to the paper in response to the reviewers, including a new Figure 3 and added text, which we believe have significantly strengthened the content, clarity, and overall story conveyed by our review.

  1. What is the evidence that FOXM1 is a driver of MPNST rather than biomarker or modulator of chemoresistance?

It is possible that FOXM1 could be a driver, biomarker, and/or modulator of MPNST progression and therapy resistance. However, there are currently too few studies to know with certainty what FOXM1 does in MPNST. We focused on FOXM1 in our review based on several intriguing observations, listed below, which support our suggestion that it may be a clinically relevant, druggable driver of disease: 1) In numerous other cancers, including many other Ras-driven cancers, FOXM1 is a well-established driver of disease; 2) FOXM1 interacts functionally with many of the central players in MPNST formation and progression. As shown in Figure 2 and Figure 4, that includes major upstream regulators / pathways like NF1-Ras-MEK-ERK, INK4a-CDK4/6-RB1, ARF, p53, and YAP, as well as downstream targets such as RB1, EZH2, and YAP; 3) One group (Yu et al.) reported that the FOXM1 gene is amplified and the FOXM1 protein overexpressed in human MPNSTs, which correlated with poor patient survival; 4) Our preliminary, unpublished studies that we cite in the review agree with those results but extend them further. We not only found significant upregulation of FOXM1 protein in MPNSTs compared to patient-matched, benign plexiform neurofibromas (PNFs), we also observed that FOXM1 is upregulated in the intermediate, pre-malignant lesion termed “ANNUBP.” ANNUBPs are not treated with any chemotherapy, suggesting its increased expression is not a result of acquired drug resistance, although FOXM1 does mediate resistance to multiple therapies in other tumor types. We consider each of the above points throughout our review.

Together, the data suggest to us that FOXM1 is a likely driver of MPNSTs that may promote the benign to malignant transformation of ANNUBPs into MPNSTs. The work by Yu et al. suggest it may be a relevant prognostic biomarker of MPNSTs although more work using larger sample sizes with associated survival information is certainly needed to verify that possibility. As we convey in the manuscript, there is much more to be learned about FOXM1 in MPNST, but there is tantalizing evidence suggesting it could be a relevant target for treating MPNSTs. 

  1. There more than dozen of known FOXM1 inhibitors. What is the logic to describe ARF peptide that fails trials or proteasome inhibitors that non-specifically inhibit FOXM1.

The ARF peptide is a relevant topic for MPNST specifically because loss of ARF occurs in the majority of these cancers and studies in mice show its genetic deletion leads to ANNUBP and MPNST development.  The use of ARF peptides as inhibitors of FOXM1 was only mentioned briefly, however, because as we explained the peptides are not suitable for clinical use. We chose to describe thiostrepton since it is the most highly cited inhibitor of FOXM1. We state that it can indirectly act against FOXM1 by inhibiting the proteasome, nonetheless there is evidence it also directly binds and inhibits FOXM1 based on in vitro assays, described in more detail in the response to point #3. That said, we appreciate the reviewer’s comment and think including more information about other FOXM1 inhibitors is an excellent suggestion. We updated this section of the review accordingly.

  1. The notion that Thiostrepton directly binds to FOXM1 and inhibits its activity is questionable. The problem is that all proteasome inhbitors inhibit FOXM1 via activation of HSP70. But is was never shown that different proteasome inhibitors with different structures bind directly to FOXM1. Therefore, it is more probable then not that general reason for proteasome inhibitors to inhibit FOXM1 is a common up-regulation of HSP70.

We wrote that thiostrepton directly binds FOXM1 and inhibits its activity based on published data reported in a 2011 Nature Chemistry paper titled, “The transcription factor FOXM1 is a cellular target of the natural product thiostrepton”. Hegde et al. measured the binding of biotinylated thiostrepton to purified GST-tagged FOXM1b and FOXM1c through streptavidin pull-down assays and western blotting as well as isothermal titration calorimetry. Binding affinities were defined. These data provide evidence for direct binding of the drug to both FOXM1 isoforms. The authors also showed decreased transcriptional activity of FOXM1 following drug treatment in cells by measuring its endogenous gene targets, which they assessed at an early 4-hour time point when the levels of FOXM1 protein were not reduced by the drug. The intent was to dissociate effects of the drug on FOXM1 versus the proteasome. In agreement with the reviewer’s concern, we presented these published results while acknowledging that clinical use of thiostrepton is limited not only by its poor pharmacokinetic properties but also its lack of specificity. We wrote that it has other important targets such as the proteasome and mitochondrial translation machinery.

Reviewer 2 Report

Comments:

1. Compared to MEK, only few FOXM1 and CDK4/6 studies on MPNST, do authors have new findings on Foxm1- and/or CDK4/6-mediated MPNST?

2.Please add post-translational modifications on FOXM1, CDK4/6, and MEK including phosphorylation, acetylation, and SUMOylation.

3. It would be better If uses full name of MPNST on title.

Author Response

We thank the reviewers for their time and effort reviewing our manuscript, as well as their constructive comments and questions. Below is a point-by-point response to each comment. We made important revisions to the paper in response to the reviewers, including a new Figure 3 and added text, which we believe have significantly strengthened the content, clarity, and overall story conveyed by our review.

  1. Compared to MEK, only few FOXM1 and CDK4/6 studies on MPNST, do authors have new findings on Foxm1- and/or CDK4/6-mediated MPNST?

Yes, there are many published studies of MEK in MPNST, relatively few on CDK4/6, and even fewer on FOXM1. Unfortunately, MPNST cell lines and PDXs are either resistant to MEK inhibitors when used alone or the drugs fail to achieve maximal efficacy. Consequently, MEK inhibitor monotherapies are not considered relevant for treating MPNSTs. While studies of cyclin D-CDK4/6 kinases are rather limited in the MPNST field, there are numerous papers demonstrating loss of their negative regulators, p16INK4a and p15INK4b, in patient MPNSTs. Those studies support findings from us and others that CDK4/6 is hyperactivated in patient tumors. Most recently, we published two studies in Clinical Cancer Research (Kohlmeyer et al, 2020 and 2023) establishing the value of targeting CDK4/6 and MEK together in MPNSTs. The 2020 paper showed elevated CDK4/6 activity in MPNSTs and decent effects of CDK4/6 inhibitor monotherapies at suppressing MPNST growth in vitro and in vivo; however, the effects of CDK4/6 inhibitors alone were transient due to the rapid outgrowth of drug resistant tumors. Combining inhibitors of CDK4/6 and CDK2 together was explored but was associated with unwanted toxicity. Because MEK activation is a well-known mechanism of resistance to CDK4/6 inhibitor therapy, we then examined dual inhibition of CDK4/6 and MEK in MPNSTs (2023 study). We found that CDK4/6-MEK inhibitor combination therapy is much more effective at killing MPNSTs than either monotherapy, that it causes changes in intratumoral immune cells, and that it sensitizes the tumors to immune checkpoint blockade using anti-PD-L1 antibodies.

While we refer to our preliminary FOXM1 findings in this review, specifically examining FOXM1 protein expression in human MPNSTs versus precursor PNF and ANNUBP lesions, we have not yet published those data. We expect that thesis research by Ms Voigt, the first author of this review, over the next few years will provide much more insight into FOXM1’s role in MPNSTs. We have used the platform provided by this special issue on “Novel therapeutic targets in cancers” to raise the exciting possibility that FOXM1 represents a new, druggable target in MPNST. The review lays out our rationale supporting that notion. We discuss the known functional connections between FOXM1 with MEK and CDK4/6 pathways in other tumors and explain why we believe FOXM1 may be an unappreciated, central player in MPNST pathogenesis whose inhibition, along with targeting of MEK and CDK4/6, may provide highly effective therapies for MPNST patients.

  1. Please add post-translational modifications on FOXM1, CDK4/6, and MEK including phosphorylation, acetylation, and SUMOylation.

Thank you for that comment. FOXM1, CDK4/6, and MEK each have many different post-translational modifications of functional significance. We respectfully believe that describing all of those modifications on all 3 proteins is not germane to the discussion and would detract from the focus of this review. However, we appreciate your recommendation since the direct effects of MEK-ERK and CDK4/6 kinases on FOXM1 phosphorylation should be more clearly conveyed. Therefore, we now include a new Figure 3 that details the phosphorylation sites regulated by cyclin D-CDK4/6 and MEK-ERK on FOXM1. We complement the new illustration with additional text that discusses the importance of those phosphorylated residues. We also cite the Kalathil and John review from 2021 that fully describes all post-translational modifications and known regulators of FOXM1.

  1. It would be better If uses full name of MPNST on title.

Great point, thank you. We have made this change to the title.  

Round 2

Reviewer 1 Report

Paper may be published in current form.

Reviewer 2 Report

No more comments